# Pressure Sensitivity Prediction and Pressure Measurement of Fast Response Pressure-Sensitive Paint Based on Artificial Neural Network

Xianhui Liao [1,2], Chunhua Wei [2], Chenglin Zuo [2], Zhisheng Gao [1], Hailin Jiang [2], Lei Liang [2,*] and Zhaoyan Li [1,*]

1   School of Computer and Software Engineering, Xihua University, Chengdu 610039, China;
    heroliao@126.com (X.L.); gzs_xihua@mail.xhu.edu.cn (Z.G.)
2   China Aerodynamics Research and Development Center, Mianyang 621000, China;
    wch410204603@126.com (C.W.); zuochenglin@cardc.cn (C.Z.); jianghailin1008@163.com (H.J.)
*   Correspondence: skywork@163.com (L.L.); lzy@mail.xhu.edu.cn (Z.L.)

**Abstract:** The characterization of pressure-sensitive paint (PSP) is affected by many physical and chemical factors, making it is difficult to analyze the relationship between characterization and influencing factors. An artificial neural network (ANN)-based method for predicting pressure sensitivity using paint thickness and roughness was proposed in this paper. The mean absolute percentage error (MAPE) for predicting pressure sensitivity is 6.5088%. The difference of paint thickness and roughness between sample and model surface may be a source of experimental error in PSP pressure measurement tests. The Stern-Volmer coefficients A and B are strongly linked. Pressure sensitivity is approximately equal to coefficient B, so coefficient A is predicted using pressure sensitivity based on the same ANN, and the MAPE of predicting A is 2.1315%. Then, we try to calculate the pressure by using the thickness and roughness on a model to predict pressure sensitivity and Stern-Volmer coefficient A. The PSP pressure measurement test was carried out at the China Aerodynamic Research and Development Center. Using the Stern-Volmer coefficient calculated by the in situ method, the method in this paper, and the sample calibration experiment, the root mean square errors (RMSE) of the pressure are 47.4431 Pa, 63.4736 Pa, and 73.0223 Pa, respectively.

**Keywords:** pressure-sensitive paint; characterization prediction; artificial neural network; pressure measurement



## 1. Introduction

Pressure-sensitive paint (PSP) pressure measurement technology is a non-contact full-field pressure measurement method based on image processing. It can provide continuous pressure measurement data, and has the advantages of high spatial resolution, low cost, short experimental preparation time, and small flow field interference. When laser or UV LED is used as an excitation light source, an oxygen quenching reaction occurs between luminescent molecules and oxygen in the paint. The changes in fluorescence intensity of the model surface paint are converted into a pseudo-color image by CCD camera. Finally, the surface pressure distribution was obtained by computer graphics processing [1].

At present, PSP technology tends to be mature and has been applied in numerous wind tunnel pressure tests [2–6]. In harsh experimental environments, such as helicopter rotor pressure tests, the image is prone to blur, and the paint may flake off due to high-speed rotor rotation and surface temperature increase. Although algorithms such as deblurring, temperature correction, and filtering can be used to correct the surface pressure distribution results [7], it greatly increases the processing time of test data and reduces the accuracy of results. Fast response PSP is a PSP that reduces response time from milliseconds to hundreds of microseconds. The emergence of fast response PSP effectively solves the problem in the helicopter rotor pressure test [8]. The ideal PSP has high pressure sensitivity,

low temperature sensitivity, and low response time. Currently, no PSP can achieve high pressure sensitivity, low temperature sensitivity, and low response time at the same time. The factors affecting the characterization of PSP need to be further studied.

The main way to study PSP characterization is to change one of the influencing factors, such as using a different probe or binder during the spraying process, changing the ratio of certain materials, or changing the thickness of the paint. Then, calibrate the characterizations through calibration experiments, and analyze the relationship between the influencing factors and characterizations through phenomenological modeling or statistical modeling. For a given performance range, such as the pressure sensitivity of a given range, we can use the relationship between the characterizations of the paint and the influencing factors to analyze the required parameters of the paint under the target characterization, such as luminous probe concentration, paint thickness, and so on. The rapid prediction of PSP characterizations can provide guidance for studying the relationship between potential influencing factors and characterizations. As early as 2004, Peng, D. et al. found that the response time of PC-PSP was affected by the thickness of the adhesive and the oxygen diffusion rate [9]. In 2018, Jiao, L. et al. established first-order models of fast PSP coating response time, oxygen diffusion of porous polymer layer, and the reaction of oxygen with luminescent molecules [10]. The research of Hayashi, T. and Okudera T. et al. in 2021 shows that the paint characterizations are not only related to the chemical properties and ratio of substrate materials [11,12], but also the physical properties such as paint thickness and roughness in the spraying process, which affect the characterization of pressure-sensitive paint. When the paint thickness is greater than 8 μm, the problem of uneven luminous life of the fast PSP paint can be improved [13]. Many studies have shown that its characterization is affected by various chemical and physical factors. These studies show that physical factors such as thickness appear to be an extremely important factor. In calibration experiments, we found that different paint thicknesses and roughnesses lead to significant differences in the pressure sensitivity of a fast-response PSP. Therefore, in PSP pressure measurement tests, differences in paint thickness and roughness between the test model and the sample may be a large source of error. The purpose of this study was to find a way to predict pressure sensitivity using paint thickness and roughness, and to find a way to avoid PSP pressure measurement errors due to differences in paint thickness and roughness between model surfaces and samples.

Neural networks have been widely touted as solving many forecasting problems [14]. At present, an artificial neural network (ANN) has been used to approximate nonlinear functions and has achieved great success in the field of engineering technology [15–19]. ANNs need large data sets to ensure their prediction accuracy. Due to the large time cost and financial cost of PSP characterization calibration, the number of data points is less than 50 data points. Based on the limited experimental observation data, a data enhancement method is used to increase the number of data points and the dropout layer is used to further improve the prediction accuracy. In order to further evaluate the prediction accuracy of this paper, we propose a method to measure the delta wing pressure by using the prediction coefficient in the Stern-Volmer equation. The pressure distribution of the delta wing is calculated by using the coefficients predicted by the ANN in this paper, and the error of this method is discussed in the test of the pressure measurement of the delta wing in the China Aerodynamic Research and Development Center (CARDC).

## 2. Experimental Setup

### 2.1. Sample Preparation

The fast response PSP used in this paper is composed of a high concentration of ceramic particles mixed with a small amount of adhesive. Its structure is shown in Figure 1. PC-PSP uses a porous structure to enhance the diffusion of oxygen through the pores, thereby rapidly responding to pressure changes.

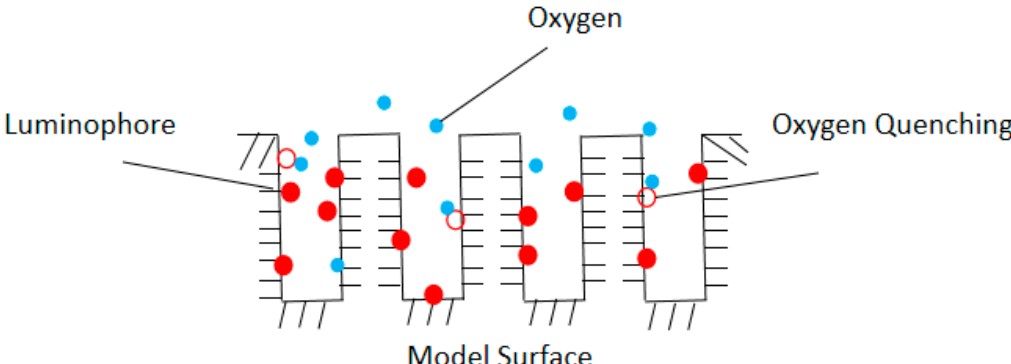

**Figure 1.** Structure schematic of the fast response PSP.

The samples are circular thin aluminum sheets. We sprayed 30 samples divided into 5 groups through the air spray method. We used the spray gun to sweep the surface of the samples at a constant speed but different sweep times: 10 times, 15 times, 20 times, 25 times, 30 times, and 35 times, to form different paint thickness and roughness. As shown in Figure 2, for PSP samples numbered 0 to 5, the sweep times increases gradually.

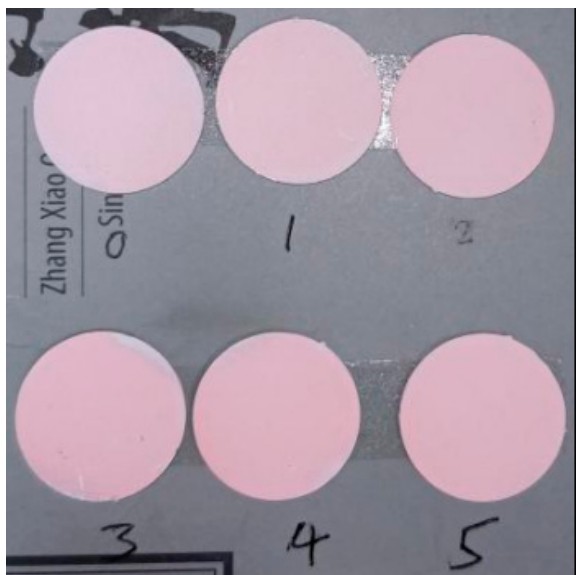

**Figure 2.** A group of samples with sweep times increased gradually.

### 2.2. Characterization Calibration of PSP

The pressure sensitivity of the PSP can be obtained with a static calibration system. The schematic of the calibration chamber is shown in Figure 3. A UV LED light source with a wavelength of 405 nm is used as the excitation light source to promote the oxygen quenching reaction. The sample is placed on the heating/cooling table, and the pressure and temperature can be changed by air source and liquid nitrogen. Pressure and temperature sensors are used to accurately read the pressure and temperature in the chamber. Pressure sensitivity is calibrated by changing the pressure under constant temperature conditions.

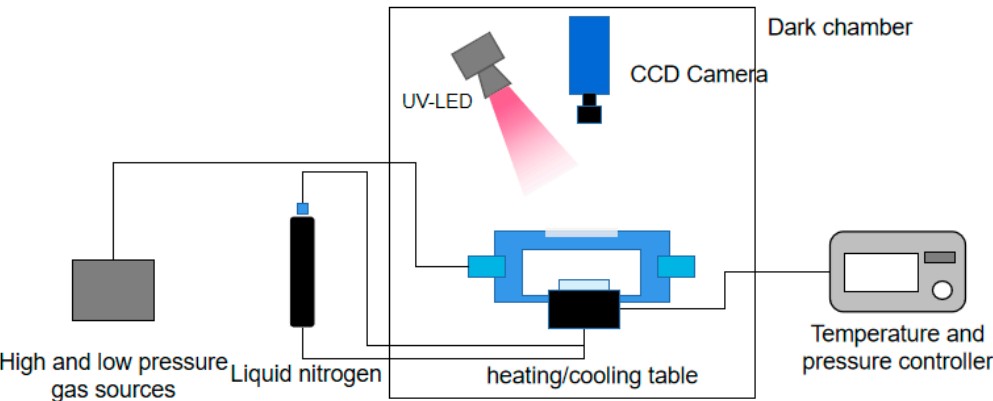

**Figure 3.** Schematic of the PSP static calibration system.

The pressure and luminescent intensity can be modeled by a simplified form of the Stern-Volmer equation, which can be expressed as:

$$\frac{I_{ref}}{I} = A(T) + B(T)\frac{P}{P_{ref}} \tag{1}$$

where $I_{ref}$ and $P_{ref}$ are the luminescent intensity and the air pressure at a reference condition, respectively. The temperature-dependent coefficients $A$ and $B$ are determined through calibration experiments. During the pressure calibration, we found that there seems to be a strong correlation between coefficients $A$ and $B$. The changes of coefficients $A$ and $B$ of the 30 samples are shown in Figure 4, and they change simultaneously and regularly.

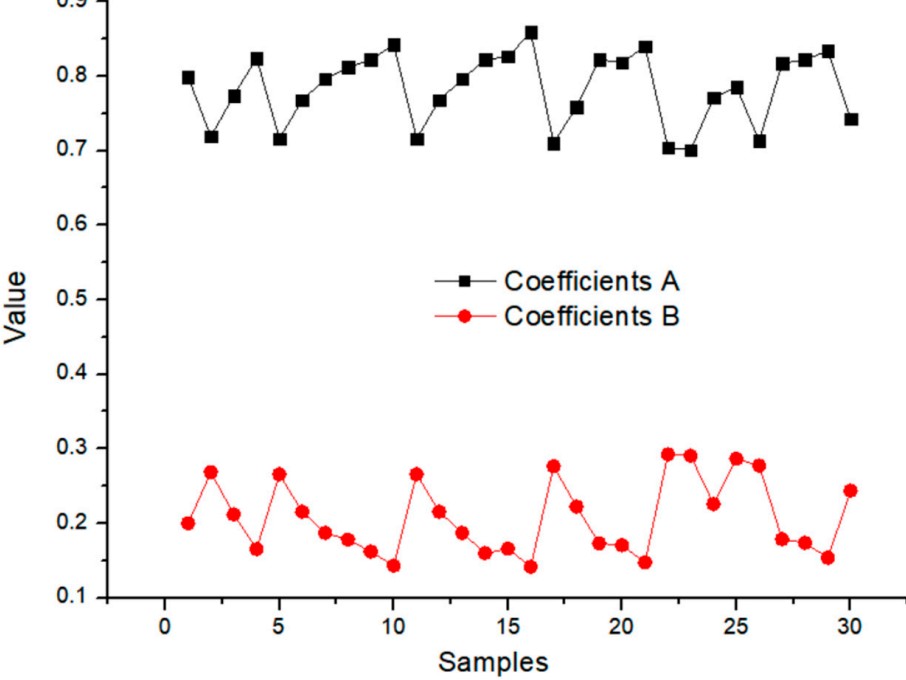

**Figure 4.** Changes of coefficients A and B of 30 samples in pressure calibration.

The pressure sensitivity is approximately equal to coefficient B. So, we try to predict the pressure sensitivity using the paint thickness and roughness of the model surface based on the ANN, then use the pressure sensitivity to predict coefficient A. We substitute the values of coefficient A and pressure sensitivity (coefficient B) into the Stern-Volmer equation

to calculate the pressure. In this way, we can avoid the problem of paint thickness and roughness differences between samples and models in PSP pressure measurement tests.

### 2.3. Dataset

Two datasets were prepared in this work. The first dataset $D_1$ consists of paint thickness, roughness, and pressure sensitivity. There are 30 pieces of data in the dataset, including training set, validation set, and test set. We selected 5 pieces of data as test set, and the remaining 25 pieces of data were randomly divided into training set and validation set. The composition of $D_1$ is shown in Table 1. Table 2 lists the maximum, minimum, and average of each element in this dataset.

**Table 1.** The composition of $D_1$.

| Sample No. | Inputs | | Outputs |
|:---:|:---:|:---:|:---:|
| | Thickness (μm) | Roughness (μm) | Pressure Sensitivity (%/kPa) |
| 1 | 15.5 | 3.27 | 0.600 |
| 2 | 23.0 | 0.90 | 0.709 |
| ... | ... | ... | ... |
| 29 | 56.6 | 1.03 | 0.767 |
| 30 | 90.6 | 0.28 | 0.774 |

**Table 2.** The maximum, minimum, and average of the elements in $D_1$.

| Component | Minimum | Maximum | Average |
|:---:|:---:|:---:|:---:|
| Thickness (μm) | 15.5 | 90.6 | 36.08 |
| Roughness (μm) | 0.38 | 4.22 | 1.24 |
| Pressure Sensitivity (%/kPa) | 0.596 | 0.815 | 0.721 |

The second dataset $D_2$ consists of pressure sensitivity and coefficient A of the same 30 samples. It is divided into training set, validation set, and test set in the same way as $D_1$. The composition of $D_2$ is shown in Table 3. Table 4 lists the maximum, minimum, and average of each element in this dataset. The ANN used to predict pressure sensitivity is denoted as $N_1$.

**Table 3.** The composition of $D_2$.

| Sample No. | Inputs | Outputs |
|:---:|:---:|:---:|
| | Pressure Sensitivity (%/kPa) | A |
| 1 | 0.600 | 0.29263 |
| 2 | 0.709 | 0.26869 |
| ... | ... | ... |
| 29 | 0.767 | 0.16209 |
| 30 | 0.774 | 0.14736 |

**Table 4.** Distribution of elements in $D_2$.

| Component | Minimum | Maximum | Average |
|:---:|:---:|:---:|:---:|
| Pressure Sensitivity (%/kPa) | 0.596 | 0.815 | 0.721 |
| $A_{pressure}$ | 0.14151 | 0.29263 | 0.2082 |

### 2.4. Pressure Measurement Test

The delta wing pressure measurement test was completed in the $\varnothing$ 0.7 m wind tunnel of CARDC. The setup of the delta wing pressure measurement test is shown in Figure 5. There are two CCD cameras near the shooting place. The fast response PSP is sprayed on the upper part of the delta wing, which is excited by a UV LED and fluoresces. First, we turned off the UV LED to create a completely dark environment. Then, 10 background images were captured using the same camera settings as in the experiment and the average of these images is called $I_{dark}$. We collected 10 images under reference and wind-on conditions and subtracted $I_{dark}$ from the corresponding raw images. Then, the average values of the images collected under these two conditions are denoted as I and $I_{ref}$, respectively. The ANN used to predict coefficient A is denoted as $N_2$.

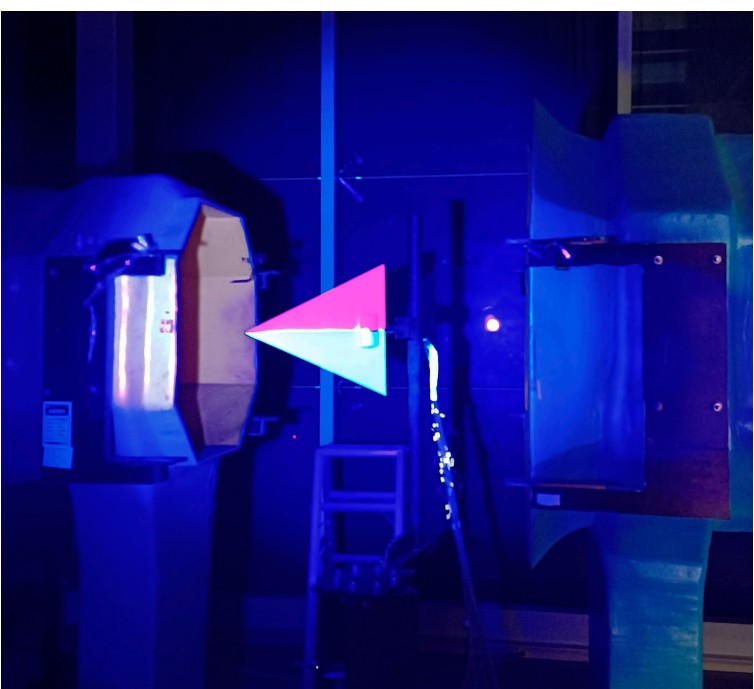

**Figure 5.** Experimental setup of the delta wing.

We sprayed the fast response PSP on a circular thin aluminum sheet to make the sample, and calibrated coefficients A and B through the static calibration system. The true value of the pressure is obtained by an electronic-scanned pressure sensor. The thickness gauge and surface roughness measuring instrument were used to measure the paint thickness and roughness on the surface of the delta wing. Then, the paint thickness and roughness were used to predict the pressure sensitivity coefficient through $N_1$, and finally, the predicted pressure sensitivity was used to predict coefficient A through $N_2$.

According to Equation (1), we can calculate the pressure of the delta wing by the following equation:

$$P = \frac{[\frac{I_{ref}}{I} - B(T)]}{A(T)} \times P_{ref} \tag{2}$$

Using the in situ calibration method [20], the calibration experiment, and the method in this paper, three groups of coefficients A and B were obtained, and Equation (2) was used to calculate the pressure. The feasibility of the method proposed in this paper is evaluated by comparing the root mean square error (RMSE) of the pressure calculated by these three methods.

### 3. Neural Network Design

*3.1. Structure*

Generally, there exist many neural network methods or types; the back propagation (BP) neural network is mature both in network theory and performance. Its outstanding advantages are strong nonlinear mapping ability and flexible network structure. The schematic of the BP neural network we used is shown in Figure 6. The neural network consists of a fully connected layer, including an input layer, an output layer, and a hidden layer. In theory, the more hidden layers a neural network has, the stronger the fitting function will be. However, deeper layers may lead to overfitting and increase the difficulty of training, thus making the model difficult to converge. We chose four hidden layers with eight neurons in each hidden layer. The ReLU function serves as the activation function of the hidden layer. In this work, we have two neural network models, $N_1$ and $N_2$, which have the same structure except for the input layer and the output layer neurons. For $N_1$, paint thickness and roughness are considered as inputs to the neural network model, while pressure sensitivity is considered as outputs for the neural network model. For $N_2$, the pressure sensitivity is the input and coefficient A is the output.

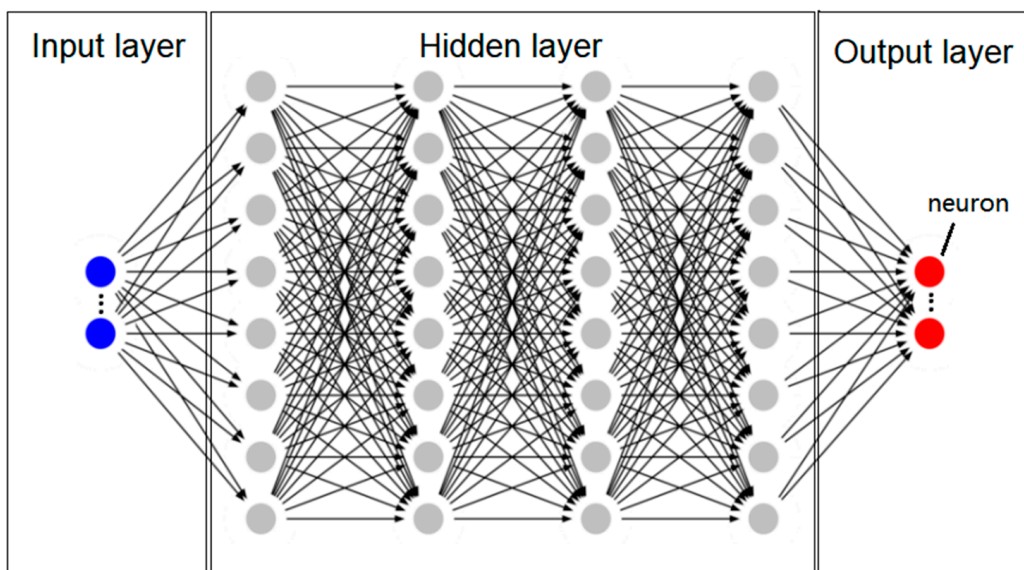

**Figure 6.** Schematic of the ANN.

*3.2. Data Augmentation*

The schematic of the data augmentation method is shown in Figure 7. The method randomly generates augmentation data points for the calibrated experimental data points within a certain confidence interval. The 30 pieces of data in $D_1$ are denoted as $A_o$ and the model trained with $A_o$ is denoted as $M_o$. Each calibration data point randomly generates 1 augmentation data point to form a new dataset, denoted as $A_1$. We call the number of augmentation data points the data augmentation factor. The same method is used to increase the data augmentation factor to 10, 50, 100, 500, and 1000; those datasets are denoted as $A_{10}$, $A_{50}$, $A_{100}$, $A_{500}$, and $A_{1000}$, respectively. The ANNs trained with the above augmentation datasets are denoted as $M_1$, $M_{10}$, $M_{50}$, $M_{100}$, $M_{500}$, and $M_{1000}$, respectively. The corresponding relationship between the data augmentation factor, dataset, and the corresponding neural network model is shown in Table 5. This method is not applied to dataset $D_2$ due to the simplicity of the prediction task.

**Figure 7.** Schematic of the data augmentation method; the data augmentation factor is 10.

**Table 5.** Dataset and corresponding neural network model.

| Data Augmentation Factor | Dataset | Model |
|:---:|:---:|:---:|
| \ | $A_o$ | $M_o$ |
| 1 | $A_1$ | $M_1$ |
| 10 | $A_{10}$ | $M_{10}$ |
| 50 | $A_{50}$ | $M_{50}$ |
| 100 | $A_{100}$ | $M_{100}$ |
| 500 | $A_{500}$ | $M_{500}$ |
| 1000 | $A_{1000}$ | $M_{1000}$ |

*3.3. Dropout*

In the model of machine learning, if the model has too many parameters and too few training samples, the trained model easily produces the overfitting phenomenon. In order to prevent overfitting in the training phase, the neurons were randomly removed. For each layer in a dense (or fully connected) network, a probability p of dropout is given. In each iteration, the probability that each neuron is removed is p. The paper by Hinton et al. suggests an input layer dropout probability of "$p = 0.2$" and a hidden layer dropout probability of "$p = 0.5$" [21]. Obviously, there is interest in the output layer, which is a prediction. So, we do not apply dropout at the output layer. The schematic of the dropout method is shown in Figure 8. To prevent overfitting under the large data augmentation factor, the dropout method is applied when the data augmentation factor is 10, 50, 100, 500, and 1000.

Mathematically, it is said that the drop probability of each neuron follows a Bernoulli distribution with probability p. Thus, an element-wise operation is performed on the neuron vector (layer) with a mask, where each element is a random variable following a Bernoulli distribution. The forward propagation of a neural network without dropout is calculated as follows:

$$y = f(Wx) \tag{3}$$

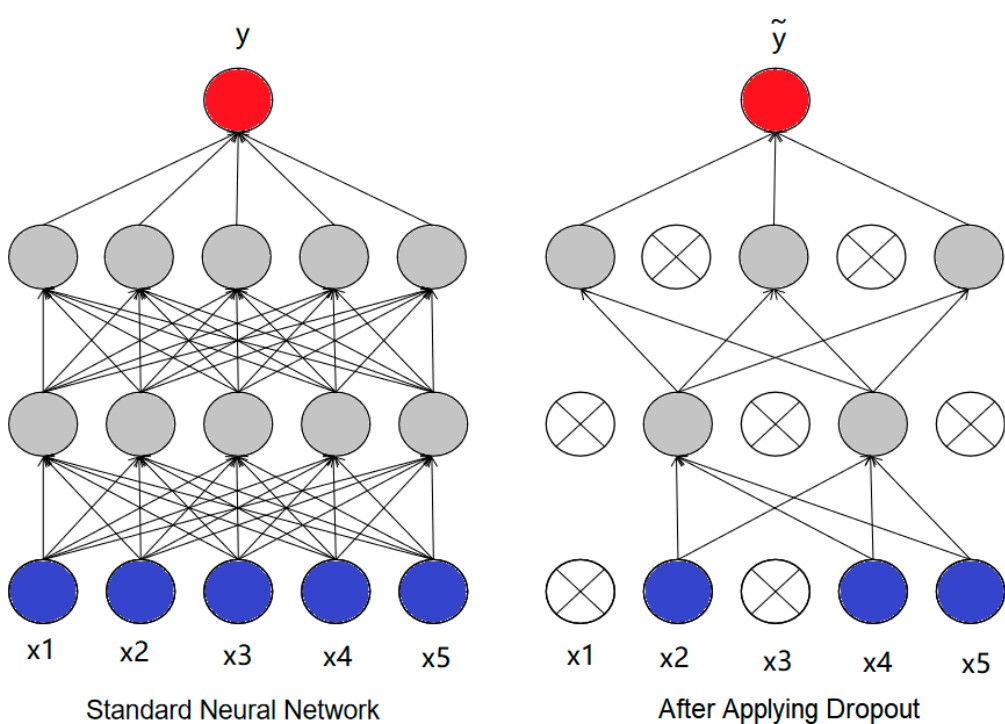

**Figure 8.** Schematic of the dropout method.

The forward propagation of a neural network with dropout is calculated as follows:

$$\widetilde{y} = f(Wx) \circ m, m_i \sim Bernoulli(p) \tag{4}$$

where $y$ is the result obtained after weighting the output $x$ of each neuron in the previous layer by weight $W$, and $m$ represents a vector consisting of multiple independent variables following the same Bernoulli distribution. The output $y$ is transformed into $\widetilde{y}$ by dropout.

*3.4. Loss Function*

The loss function is used to represent the difference between the predicted result y* and the true value y. In the process of neural network training, the loss function is continuously reduced by changing all the parameters in the neural network, so as to train the neural network model with higher accuracy. Mean absolute percentage error (MAPE) is expressed as follows and used to evaluate the model:

$$MAPE = \frac{100}{n} \sum_{i=1}^{n} \frac{\left| y_i - y_i^* \right|}{y_i} \% \tag{5}$$

*3.5. Training*

Typically, the dataset is divided into the training set and the validation set in a ratio of 8:2. However, in order to make the training set as large as possible to ensure the prediction accuracy, the ratio is 9:1 in this work because the dataset is very small. $N_1$ and $N_2$ are trained in the same way. The initial learning rate is $10^{-4}$. The Adam algorithm was used to optimize the neural network, and network parameters affecting model training and model output were updated and obtained. The ANN was trained 40,000 rounds to make it close to or reach the optimal value.

## 4. Results and Discussion

### 4.1. Results of $N_1$

Table 6 shows the prediction error obtained by training $N_1$ with $A_o$, $A_1$, $A_{10}$, $A_{50}$, $A_{100}$, $A_{500}$, and $A_{1000}$. We used MAPE to evaluate the error between the predicted and true values for the five entries in the test set. Each ANN model is trained 40,000 times separately. Since the initial coefficients of the network are random, the final evaluation result is the average of the evaluation results after five training sessions.

**Table 6.** Prediction error of $N_1$.

| Model | MAPE (%) |
|---|---|
| $M_o$ | 9.1749 |
| $M_1$ | 10.2517 |
| $M_{10}$ | 9.2730 |
| $M_{50}$ | 8.8186 |
| $M_{100}$ | 9.1405 |
| $M_{500}$ | 9.1208 |
| $M_{1000}$ | 9.0245 |

Without using the data augmentation method, the MAPE of predicting pressure sensitivity is about 9.1749%. After using the data augmentation method, the lowest MAPE is 8.8186%, and the data augmentation factor is 50. When the data augmentation factor is 100, the prediction error is 9.1405%, which is greater than $M_{50}$. With the increase of the data augmentation factor, the change of MAPE is negligible. This may be due to the limited features extracted by the neural network and the result of overfitting after neural network training when the original data set is small and the data augmentation factor is too large.

In order to reduce the effect of overfitting in training, we try to apply the dropout method to $M_{10}$, $M_{50}$, $M_{100}$, $M_{500}$, and $M_{1000}$. A dropout layer is added after each hidden layer and the new models are denoted as $M_{10\_dropout}$, $M_{50\_dropout}$, $M_{100\_dropout}$, $M_{500\_dropout}$, and $M_{1000\_dropout}$. The MAPE and improvements of the above models are shown in Table 7.

**Table 7.** Prediction error of $M_{10}$, $M_{50}$, $M_{100}$, $M_{500}$, $M_{1000}$, $M_{10\_dropout}$, $M_{50\_dropout}$, $M_{100\_dropout}$, $M_{500\_dropout}$, and $M_{1000\_dropout}$.

| Model | MAPE (%) | Improvement (%) |
|---|---|---|
| $M_{10}$ | 9.2730 | |
| $M_{10\_dropout}$ | 7.0122 | 2.2608 |
| $M_{50}$ | 8.8186 | |
| $M_{50\_dropout}$ | 6.5088 | 2.3098 |
| $M_{100}$ | 9.1405 | |
| $M_{100\_dropout}$ | 7.1127 | 2.0278 |
| $M_{500}$ | 9.1208 | |
| $M_{500\_dropout}$ | 6.9369 | 2.1839 |
| $M_{1000}$ | 9.0245 | |
| $M_{1000\_dropout}$ | 6.8677 | 2.1568 |

In the case of the same data augmentation factor, the prediction accuracy of the neural network after applying the dropout method is improved by about 2% in each model. It is proved that the dropout method can effectively improve the prediction accuracy of pressure sensitivity under large datasets. Among the neural networks without dropout, $M_{50}$ has the lowest MAPE of 8.8186%. Compared with $M_{50\_dropout}$ neural network model, the prediction accuracy is improved by 2.3098%. It can be seen that for the fast response PSP characterization prediction task, combining the data augmentation method with the dropout method can further improve the prediction accuracy of the neural network.

$M_{50\_dropout}$ is used to predict pressure sensitivity. The y=x function is used to indicate that the true value is equal to the predicted value. The red area indicates the 5% error band of the true values. Plotting the distribution of the predicted values of pressure sensitivity within the 5% error band on the training set and test set is shown in Figures 9 and 10, respectively. On the training set there are 1250 points, and most of the predicted values of pressure sensitivity are distributed around the error range of the true values. The MAPE of predicting pressure sensitivity on the test set was 6.5088%, slightly higher than the 5.5251% on the training set.

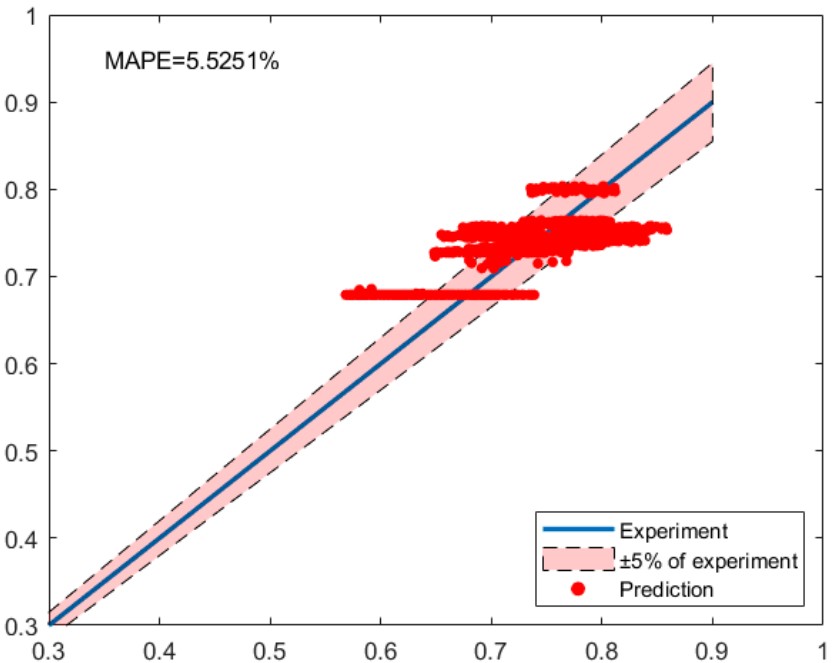

**Figure 9.** Distribution of pressure sensitivity prediction points in the 5% error band on the training set.

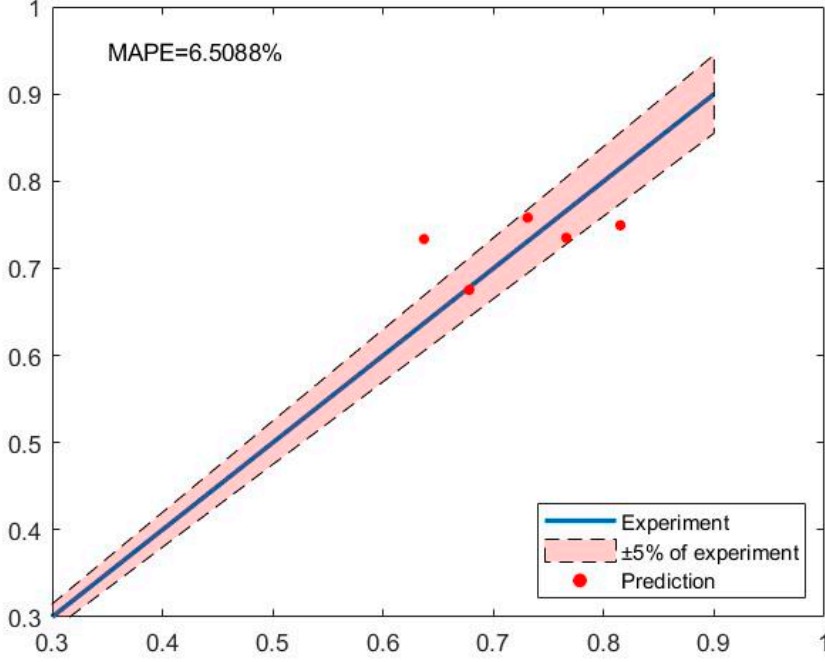

**Figure 10.** Distribution of pressure sensitivity prediction points in the 5% error band on the test set.

*4.2. Result of $N_2$*

We trained $N_2$ using the dataset $D_2$. The predicted results and experimental values of coefficient A are shown in Table 8. The absolute error of prediction is less than 0.001, and the MAPE between the prediction and the experimental value of the five data points in the test set is 2.1315%. The distribution of the predicted values for the 5% error band coefficient A on the training set and test set is shown in Figures 11 and 12, respectively. All the predicted points of coefficient A on the test set are distributed within the 5% error band of the true value. The MAPE of predicting coefficient A on the test set is 2.1315%, which is smaller than the 5.4576% on the training set. This may be due to the small data set.

**Table 8.** Absolute error of coefficient A.

| Sample No. | Experimental Value | Prediction | Absolute Error |
|---|---|---|---|
| 1 | 0.27744 | 0.2784 | 0.0010 |
| 2 | 0.17874 | 0.1792 | 0.0004 |
| 3 | 0.17379 | 0.1744 | 0.0006 |
| 4 | 0.15363 | 0.1629 | 0.0093 |
| 5 | 0.24381 | 0.2498 | 0.0060 |

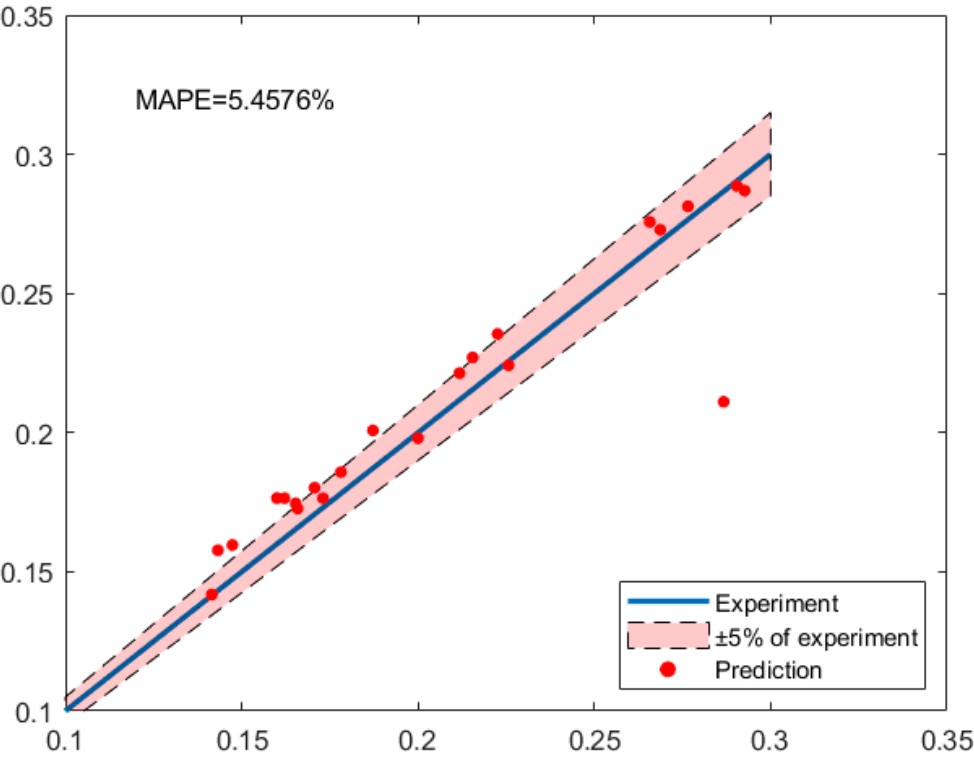

**Figure 11.** Distribution of coefficient A prediction points in the 5% error band on the training set.

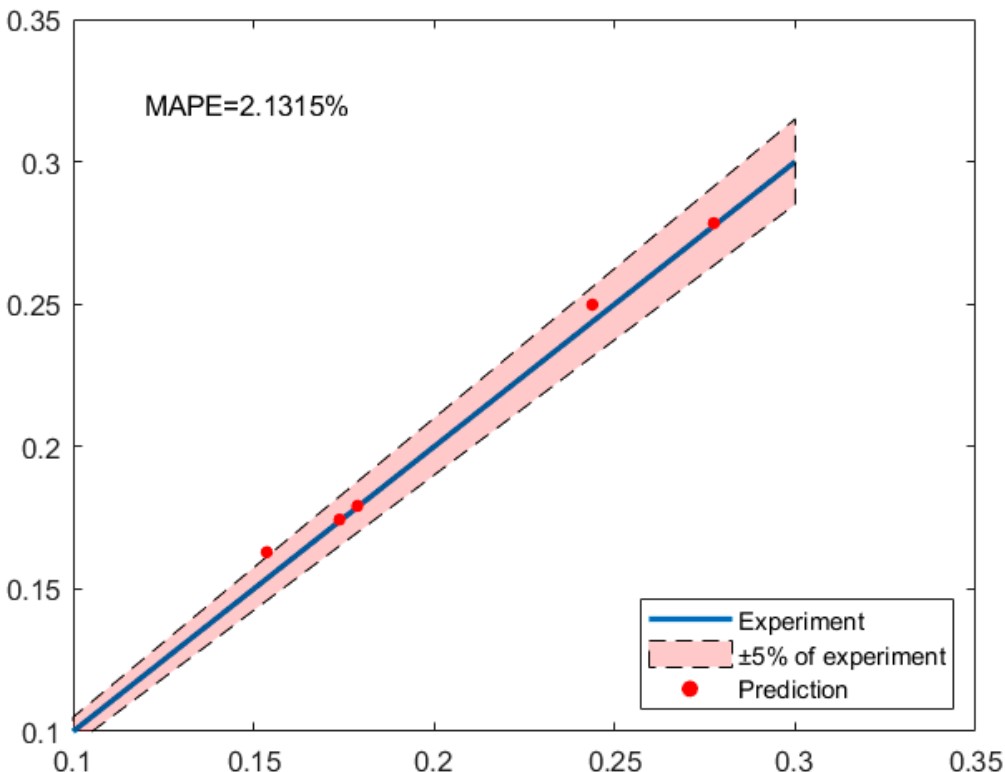

**Figure 12.** Distribution of coefficient A prediction points in the 5% error band on the test set.

*4.3. Results of Pressure Measurement Test*

During the experiment, the reference pressure $P_{ref}$ in the wind tunnel was 95,000 Pa, and the temperature in the wind tunnel changed from 294.85 K to 295.45 K after heat balance. The angle of attack of the delta wing was set to 30° and the wind speed was set to 30 m/s. The image taken under wind-on conditions is shown in Figure 13. The pressure holes in the chord direction are denoted as S1, S2, S3, S4, and S5. The global pressure distribution diagram of the delta wing calculated by the method proposed in this paper is shown in Figure 14. There is a distinct area of low pressure at the front of the delta wing. In addition, the Stern-Volmer coefficients A and B are obtained through the in situ method and calibration experiment, and the pressure measurement results are obtained according to Equation (2) and compared with the method proposed in this paper.

In order to find out whether the data augmentation method and the dropout method can improve the prediction accuracy of pressure sensitivity, we use the trained neural networks $M_o$ and $M_{50}$, which have the best effect after using these two methods, to predict pressure sensitivity. Then we use the two pressure sensitivities through $N_2$ to predict coefficient A, and finally calculate the pressure separately. We denoted the pressure obtained using these two models as method in this paper-$M_o$ and method in this paper-$M_{50}$. The pressure distributions of the 24 pressure holes from S3-1 to S5-10 obtained by using the in situ method, the method proposed in this paper-$M_o$, the method proposed in this paper-$M_{50}$, and the calibration experiment are shown in Figure 15. Using the pressure measured by the electronic-scanned pressure sensor as the true value, we evaluated the four methods using RMSE. The RMSE of the four methods is shown in Table 9. The pressure value measured by the in situ method is closest to the true value, and the RMSE of the method in this paper-$M_{50}$ is less than that of the method in this paper-$M_o$, which proves that the data augmentation method and dropout method can improve the accuracy of pressure sensitivity prediction. So, we should use the $M_{50}$ to predict the pressure sensitivity. The RMSE of the method in this paper-$M_{50}$ is greater than that of the in situ method, but less than that of the sample calibration experiment. It can be said that the accuracy of the ANN is quite accurate and worthy of attention. What needs illustration is that when

the pressure hole is located at the bottom of the delta wing, the error is very large, which may be the result of ignoring the inhomogeneity of ultraviolet light and the influence of temperature in this paper.

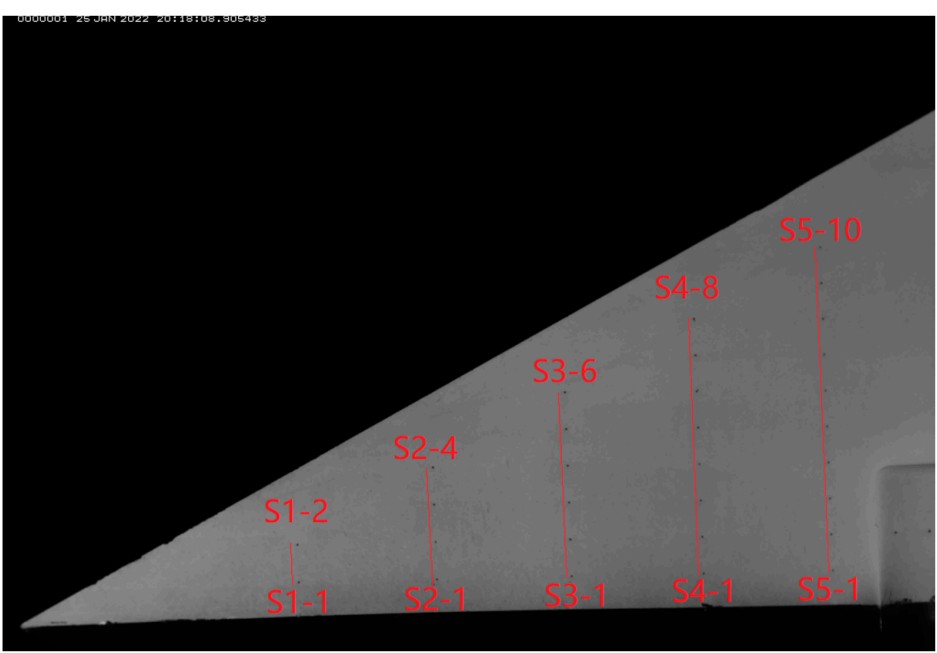

**Figure 13.** The image taken under wind-on conditions and distribution of pressure holes.

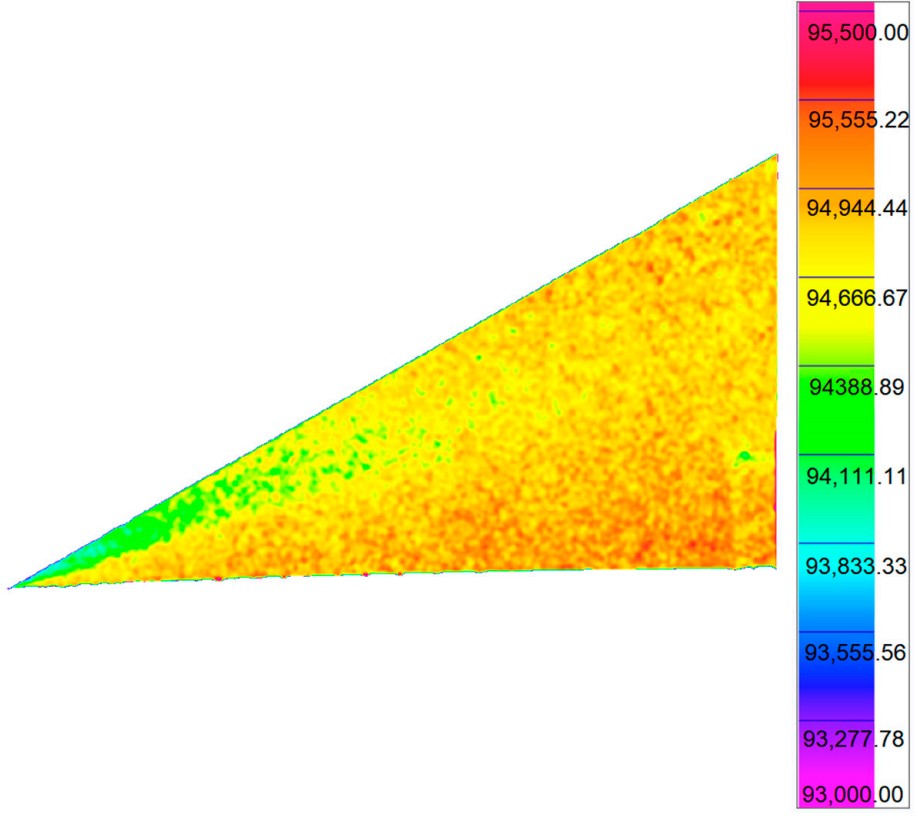

**Figure 14.** The global pressure distribution diagram of the delta wing calculated by the method proposed in this paper.

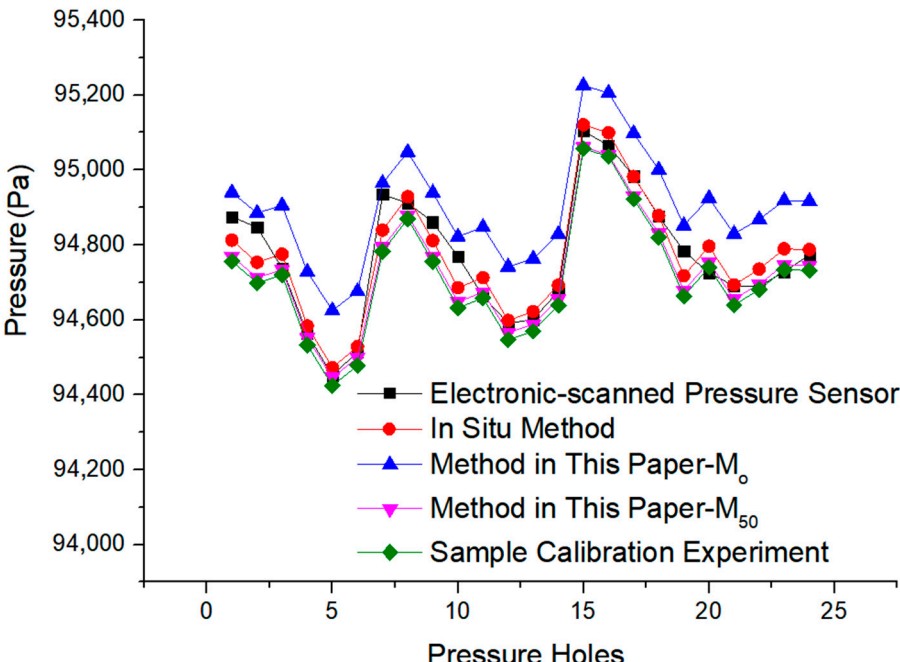

**Figure 15.** The distribution of pressure measured by electronic-scanned pressure sensor, the in situ method, the method in this paper-$M_o$, the method in this paper-$M_{50}$, and the sample calibration experiment.

**Table 9.** RMSE of pressure calculated by the in situ method, the method in this paper-$M_o$, the method in this paper-$M_{50}$, and the sample calibration experiment.

| Methods | RMSE (Pa) |
|---|---|
| In situ | 47.4431 |
| Method in this paper-$M_o$ | 63.4736 |
| Method in this paper-$M_{50}$ | 139.6155 |
| Sample calibration experiment | 73.0223 |

## 5. Conclusions

In this work, the pressure sensitivity is predicted quickly and accurately through the thickness and roughness of the paint based on an ANN. After applying the data augmentation method, its prediction error MAPE is reduced from 9.2730% on the original data set to 8.8186%. In order to prevent over-fitting in the training stage, the discarding method was adopted, and its MAPE was reduced to 6.5088%. Through the PSP pressure measurement experiment, it is further verified that the two methods can improve prediction accuracy. In terms of characterization research, this method can quickly obtain the relationship between pressure sensitivity and paint thickness and roughness. After inputting paint thickness and roughness, pressure sensitivity can be obtained. By inputting multiple sets of paint thickness and roughness into the trained ANN, and then screening the pressure sensitivity obtained from the ANN, the paint thickness and roughness required for the target pressure sensitivity can be obtained. When studying the relationship between multiple influencing factors and paint characterizations, the time cost and economic cost of calibration experiments can be reduced to a certain extent.

In the PSP pressure measurement tests, in order to avoid the pressure measurement error due to the difference in paint thickness and roughness between the model surface and the sample, we propose a new ANN-based method to measure the pressure: we use the ANN to predict coefficients A and B in the Stern-Volmer equation. We use the neural network $N_1$ to predict the pressure sensitivity through the paint thickness and the roughness of the model. The pressure sensitivity is approximately equal to B, so we use the pressure

sensitivity to predict coefficient A through N$_2$. The MAPE of prediction coefficient A is 2.1315%. According to the Stern-Volmer equation, the delta wing pressure was calculated and compared with the in situ method and the sample calibration experiment. In the delta wing pressure measurement test, the RMSE of the in situ calibration, the method in this paper, and the pressure measured by the calibration experiment are 47.4431 Pa, 63.4736 Pa, and 73.0223 Pa, respectively. The RMSE of the method in this paper is greater than the in situ method, but smaller than the sample calibration experiment, so the ANN-based method to measure pressure in the PSP pressure measurement experiment deserves attention.

**Author Contributions:** Conceptualization, C.W. and L.L.; data curation, H.J.; funding acquisition, X.L. and C.W.; methodology, X.L., C.Z., Z.G. and Z.L.; writing—original draft, X.L.; writing—review and editing, L.L. and Z.L. All authors have read and agreed to the published version of the manuscript.

**Funding:** This research was supported by the National Key R&D Program of China (Grant No. 2020YFA0405700), the National Natural Science Foundation of China (NSFC No. 12202476), and supported by the Innovation Fund of Postgraduate, Xihua University (Grant Number: YCJJ2021033).

**Institutional Review Board Statement:** Not applicable.

**Informed Consent Statement:** Not applicable.

**Data Availability Statement:** Not applicable.

**Acknowledgments:** The authors appreciate the assistance from undergraduate intern Yuning Li during dataset preparation.

**Conflicts of Interest:** The authors declare no conflict of interest.

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
