# Peer review of "Pressure Sensitivity Prediction and Pressure Measurement of Fast Response Pressure-Sensitive Paint Based on Artificial Neural Network"

_applsci, doi:10.3390/app13063504_

Round 1

Reviewer 1 Report

The authors proposed ‘An artificial neural network (ANN) based method for predicting pressure sensitivity using paint thickness and roughness’ and obtained good results and graphics. However, there are a number of issues that the authors need to look into to further enhance the paper’s presentation. 1. The use of the term ‘An artificial neural network (ANN)’ in this paper without specifying the exact type, or name of the neural network method or model used makes the entire paper presentation vague. And as such, one cannot pinpoint the contribution to knowledge made. Generally, there exist many neural network methods or types: Multilayer perceptron neural networks, Generalized regression neural networks, Radial basis function neural networks, etc. Please, specify the exact ANN name, model and algorithm used in this paper. This is lacking 2. Clearly outline at least two or more contributions to knowledge you achieved in this paper. 3. Provide a detailed related works/literature review section. This is also lacking in the paper 4. Provide a clear statement for research methodology adopted, accompanying it with a flowchart or flow diagram 5. What is the complexity of the ANN model used 6. Provide details of how the number of neurons and layers affect the precision accuracy of the ANN model

Author Response

Thanks for your comments and precious advice which are really constructive. Responses to the comments are given below, and the amendments are highlighted in red in the revised manuscript. Please see the attachment.

Reviewer 2 Report

The manuscript entitled "Pressure Sensitivity Prediction and Pressure Measurement of Fast Response Pressure-sensitive Paint Based on Artificial Neural Network” has been investigated in detail. The topic addressed in the manuscript is potentially interesting and the manuscript contains some practical meanings, however, there are some issues which should be addressed by the authors:

·         The authors stated that “The difference of paint thickness and roughness between sample and model surface may be a source of experimental error in PSP pressure measurement tests. “. Do the authors check reproducibility for experiments? How many times each experiment was repeated? Please explain in the manuscript.

·         How the best structure for ANN model has been obtained? For example, how the best number of neurons, epochs, activation functions, number of hidden layers and etc. were obtained? Please explain in the text precisely.

·         I recommend the authors to review other recently developed works.

·         The authors should clearly emphasize the contribution of the study. Please note that the up-to-date of references will contribute to the up-to-date of your manuscript. The Artificial Intelligence based studies in 2021 such as “Application of GMDH neural network technique to improve measuring precision of a simplified photon attenuation based two-phase flowmeter” could be used in the study or to indicate the contribution in the "Introduction" section.

·         Please separate the regression diagrams in Fig 9 and 10 for training and testing sets.

·         Other Errors such as MRE, RMSE, R^2 are suggested for using in the paper.

This study may be consider for publication if it is addressed in the specified problems.

Author Response

(The authors gave the same response as above.)

Round 2

Reviewer 1 Report

The authors has effected the corrections satisfactorily

Reviewer 2 Report

My recommendation is accept in present form.